# *DCDC2*-Related Ciliopathy: Report of Six Polish Patients, Novel *DCDC2* Variant, and Literature Review of Reported Cases

**DOI:** 10.3390/diagnostics13111917

**Published:** 2023-05-30

**Authors:** Patryk Lipiński, Elżbieta Ciara, Dorota Jurkiewicz, Magda Mekrouda, Joanna Cielecka-Kuszyk, Elżbieta Jurkiewicz, Rafał Płoski, Joanna Pawłowska, Irena Jankowska

**Affiliations:** 1Department of Pediatrics, Nutrition and Metabolic Diseases, The Children’s Memorial Health Institute, 04-730 Warsaw, Poland; 2Department of Medical Genetics, The Children’s Memorial Health Institute, 04-736 Warsaw, Poland; 3Department of Gastroenterology, Hepatology, Feeding Disorders and Pediatrics, The Children’s Memorial Health Institute, 04-730 Warsaw, Poland; 4Department of Pathology, The Children’s Memorial Health Institute, 04-730 Warsaw, Poland; 5Department of Diagnostic Imaging, The Children’s Memorial Health Institute, 04-730 Warsaw, Poland; 6Department of Medical Genetics, Medical University of Warsaw, 02-091 Warsaw, Poland

**Keywords:** ciliopathy, *DCDC2*, cholestasis, liver transplantation, next generation sequencing

## Abstract

Introduction: The increasing usage of NGS technology has enabled the discovery of new causal genes in ciliopathies, including the *DCDC2* gene. The aim of our study was to present the clinical, pathological and molecular report of six patients (from three unrelated families) with *DCDC2* biallelic pathogenic variants. A detailed overview of the reported patients with *DCDC2*-related disease was provided. Material and methods: A retrospective chart review of the clinical, biochemical, pathological (liver histology) and molecular features of the study group was performed. The database PubMed (MEDLINE) was searched for relevant studies. Results: All the patients presented with cholestatic jaundice and elevated GGT; the mean age was 2 months. The initial liver biopsy was performed in four children at a mean age of 3 months (age range: 2–5 months). In all of them, features of cholestasis, portal fibrosis and mild portal inflammation were observed; in three of them ductular proliferation was observed. One patient had undergone liver transplantation (LTx) at 8 years of age. At hepatectomy, a biliary-pattern cirrhosis was observed. Only one patient presented with features of renal disease. Whole exome sequencing was performed in all patients at the last follow-up visit (mean age 10 years). Three different variants (one novel) in the *DCDC2* gene were identified in the study group. With our six patients, a total of 34 patients with *DCDC2*-related hepatic ciliopathy were identified. The main clinical presentation of *DCDC2*-related ciliopathy was liver disease in the form of neonatal sclerosing cholangitis. The predominance of early and severe liver disease associated with no or mildly expressed kidney involvement was observed. Conclusions: Our findings expand the molecular spectrum of pathogenic *DCDC2* variants, provide a more accurate picture of the phenotypic expression associated with molecular changes in this gene and confirm a loss of functional behaviour as the mechanism of disease.

## 1. Introduction

The term ciliopathy is attributed to the group of disorders resulting from abnormal formation or function of the primary (immotile) cilia [1,2]. As a result of its presence in nearly all tissues and organs, the impairment of their structure/function may result in heterogeneous phenotypes, ranging from single organ involvement to multi-systemic disorders [1,2]. However, some ciliopathies have variable tissue and organ predilection. In most of them, a combined kidney (fibrocystic renal disease) and liver (congenital hepatic fibrosis and/or Caroli disease) involvement is observed; this group is also called hepatorenal fibrocystic disorders (HRFCDs) [3]. Autosomal recessive polycystic kidney disease (ARPKD) is a prototype of HRFCD, and both along with autosomal dominant polycystic kidney disease (ADPKD) represent the most common ciliopathies [3,4]. In contrast to polycystic kidney disease, nephronophthisis-related ciliopathies (NPHP-RC), including isolated (#256100) and syndromic nephronopthisis (Senior-Loken syndrome, #266900; Joubert syndrome, #213300; Meckel Gruber syndrome, #249000) constitute the second group of HRFCDs [5].

The introduction of next-generation sequencing (NGS) technology, including targeted gene panels, whole-exome sequencing (WES) or even whole-genome sequencing (WGS) have now opened promising possibilities to identify the molecular background of genetic diseases [6,7]. An increasing usage of NGS technology has also enabled the discovery of new causal genes in ciliopathies, including the *DCDC2* gene [8].

In 2015, Schueler et al. [8] provided the first report on two patients with nephronophthisis and early-onset severe hepatic fibrosis due to *DCDC2* mutations. By immunofluorescence studies, the authors showed that *DCDC2* localizes to the ciliary axoneme in the cilia of cholangiocytes and of renal epithelial cells [8]. The *DCDC2* gene was then added to the list of genes causing renal-hepatic ciliopathy. In recent years, more and more patients with isolated liver disease or combined liver and kidney disease associated with *DCDC2* pathogenic variants have been described. However, the detailed clinical presentation was not available in some of the reported cases, and thus created some confusion.

The aim of our study was to present the clinical, pathological and molecular report of six patients (from three unrelated families) with *DCDC2* biallelic pathogenic variants. A detailed overview of the reported patients with *DCDC2*-related disease was provided.

## 2. Materials and Methods

### 2.1. Patients

The presentation at the time of diagnosis and detailed follow-up were described. Informed and written consent was obtained from the parents of patients. Ethical approval was obtained from the Children’s Memorial Health Institute Bioethical Committee, Warsaw, Poland.

A retrospective chart review of patients’ medical records concerning the biochemical (platelets, serum transaminases, total and direct serum bilirubin, gamma-glutamyl transpeptidase, internal normalized ratio, serum bile acids), pathological (microscopical examination of liver biopsy specimens) and molecular (targeted to cholestatic liver disease genes) were collected.

### 2.2. Liver Biopsy

The liver biopsy specimens, between 1.0 and 1.5 cm in length, were fixed in 4% formalin and stained by hematoxillin and eosin, the PAS method (periodic acid + Schiff reagent) and PAS method after diastase digestion, Azan method, reticulin impregnation and immunohistochemistry for cytokeratin detection was applied. To assess the histological activity of microscopical changes, the following categories of lesions were considered retrospectively: presence of inflammatory infiltrates in the portal spaces and lobules with or without piecemeal necrosis, degree of fibrosis, lobular disarray, rosette formations, proliferation of bile ducts and ductules with or without ductitis, lobular necrosis, hepatocyte giant cell transformation, steatosis and degenerative changes in the hepatocytes, canalicular bile plugs, cholestasis in the hepatocytes and in bile ducts, extramedullary hematopoesis.

### 2.3. Molecular Analysis

Genomic DNA was extracted from peripheral blood samples of the patients. Whole exome sequencing (WES) was performed in all the patients after the negative result of next-generation sequencing (NGS) of the targeted gene panel, created by The Children’s Memorial Health Institute for the simultaneous sequencing of more than 1000 clinically relevant genes including 53 items related to cholestatic liver disorders.

The nomenclature of molecular variants follows the Human Genome Variation Society guidelines (HGVS, http://varnomen.hgvs.org/) (accessed on 1 March 2023) using human cDNA sequencing of appropriate genes followed the Human Gene Mutation Database (HGMD, http://www.hgmd.cf.ac.uk) (accessed on 1 March 2023). American College of Medical Genetics and Genomics (ACMG) and the Association for Molecular Pathology (AMP) standardized guidelines for sequence-level variant classification in Mendelian diseases were used.

### 2.4. Literature Search

The database PubMed (MEDLINE) was searched for relevant studies on 10 March 2023. The following keyword-based strategy was used: (“neonatal sclerosing cholangitis” OR “NCS” AND (“*DCDC2*”). All studies, letters, and abstracts that contained sufficient data were included. Available data regarding the clinical presentation, biochemical and histological features, treatment, and outcome, as well as molecular characteristics, were extracted.

## 3. Results

### 3.1. Patients’ Presentation

Six Polish patients, including three males and three females, from three unrelated families were identified; see Table 1. The age at first clinical presentation was about 2 months (age range: 2nd day of life—3 months). All the patients presented with cholestatic jaundice and elevated GGT; in two (33%) of them, the presence of acholic stools in the neonatal period was observed.

The initial liver biopsy was performed in four children with a mean age of 3 months (age range: 2–5 months). In all of them, features of cholestasis, portal fibrosis and mild portal inflammation were observed; in three of them ductular proliferation was observed; see Table 2 and Figure 1. Two patients (Pt 1 and 6) were biopsied at later time points (3.5 and 2.5 years, respectively). Their specimens demonstrated a paucity of interlobular bile ducts and severe fibrosis.

One patient (Pt 1) had undergone liver transplantation (LTx) at 6 years of age. At hepatectomy, biliary-pattern cirrhosis was observed, see Table 2 and Figure 1**.**

Only one patient (Pt 1) of three patients who underwent magnetic resonance cholangiopancreatography (MRCP) presented with features of intrahepatic cholangiopathy; see Figure 2.

Only one patient (Pt 2) presented with features of renal disease; this patient was diagnosed with right vesico-ureteral reflux and persistent urachus (Table 1).

Whole exome sequencing (WES) was performed in all patients at the last follow-up visit (2021–2022). The mean age at *DCDC2*-related ciliopathy diagnosis was 10 years (age range 9 months—24 years).

Three different variants in the *DCDC2* gene were identified in the study group (Table 1). All patients were compound heterozygotes, except one homozygote (Pt 6). Two small deletions leading to frameshift and premature termination codon and one substitution in acceptor splice site were detected. The deletion NM_016356.5: c.256del, NP_057440.2:p.(Tyr86Thrfs*17) was not reported previously, but was detected in trans with another known pathogenic variant (PM3). This variant was absent in null control chromosomes in GnomAD project (PM2_supporting). No computational evidence and in-silico tools support a deleterious effect, but this site is strongly conserved according to phyloP (score = 9.182) so is predicted a pathogenic outcome of this variant (PP3_supporting). It is expected, this frame shift (null) variant in gene *DCDC2* cause loss of protein function due to NMD. Loss of function is a known mechanism of disease (PVS1). To date no clinical significance assessments have been submitted for this variant to ClinVar or HGMD database. According to ACMG/AMP classification variant received 12 points, which corresponds to the status: Pathogenic.

### 3.2. Literature Review

A total number of 28 patients (from 24 families) affected with *DCDC2*-related ciliopathy were identified [8,9,10,11,12,13,14,15,16,17]. A detailed clinical, biochemical, pathological and molecular characterisation was provided in Table 3.

The most common presenting symptom was cholestatic jaundice and the mean age at presentation was 5 weeks (age range: 1 week—9 months). In most of them, an elevated GGT activity was reported. It is to be noted that a detailed clinical presentation was not available in some of the reported cases.

Seventeen patients (60%) required LTx at a mean age of 11 years old (range 8 months—25 years), while in two of them, a combined liver and kidney transplantation (LKTx) was performed.

Twenty-two patients had undergone liver biopsy, while in seven of them there were 2twobiopsies performed (including five histopathologic examinations of the explanted liver). In 13 patients, the liver biopsy was performed in the first 12 months of age (mean age: 5 months); in all of them features of cholestasis as well as fibrosis were observed.

Eleven patients presented with features of renal disease, including nephronophthisis (3/11), renal failure/insufficiency (4/11), and congenital anomalies (2/11).

Three patients (Pt 14, 21, 22) presented with psychomotor delay, while one of them had also microcephaly. One patient (Pt 16) presented with unilateral sensorineural hearing loss and brain imaging abnormalities. One patient (Pt 15) presented with internal carotid artery aneurysms.

Seventeen pathogenic or likely-pathogenic NSC-related variants were reported among patients in the literature, including five (29%) splice-site, four (24%) frame-shift indels, four (24%) missense and three (18%) nonsense and one in-frame deletion.

## 4. Discussion

The *DCDC2* (#605755) gene is located on the 6p22.3 chromosome and encodes a ciliary protein with two doublecortin domains, known to bind tubulin and enhance microtubule polimerization [18]. According to Online Mendelian Inheritance in Man (OMIM), three clinical autosomal recessive phenotypes are associated with *DCDC2* pathogenic variants, including deafness type 66 (DNFB-66, #610212), nephronophthisis type 19 (NPHP19, #616217) and neonatal sclerosing cholangitis (NSC, #617394) [18].

Neonatal sclerosing cholangitis (NSC) is a rare and severe form of cholangiopathy. It was first reported in 1987 in eight children presenting in the neonatal period with cholestatic jaundice, hepatosplenomegaly, acholic stools, and elevated GGT, raising a clinical suspicion of biliary atresia (BA) [19]. Radiologic abnormalities of the intrahepatic and (in most of them) extrahepatic bile ducts, consistent with sclerosing cholangitis were observed. Jaundice subsided within several months of life; however, all the patients developed the clinical, biochemical and histological features of fibrosis or cirrhosis in the infancy [19].

In 2002 and subsequently 2014, French Authors described the first patients with neonatal cholestasis mimicking BA and a typical pattern of NSC in cholangiography [19,20]. Additionally, all the patients presented with diffuse ichthyosis. Molecular analyses identified *CLDN1* (coding for claudin-1) defects, and the authors assigned a syndrome associating NSC and ichthyosis (#607626, NISCH syndrome) [20,21].

The first report of *DCDC2* mutations (biallelic missense mutations or in-frame deletion) in NSC was provided by Girard et al. in a whole exome sequencing (WES) study in four patients from two unrelated consanguineous families [9]. Renal involvement was expressed heterogeneously, from its absence to congenital anomalies of the kidney and urinary tract (CAKUT) or development of renal insufficiency just after liver transplantation (LTx) [9]. In a study of Grammatikopoulos et al., 4 of 13 patients with NSC were found affected with mutations in *DCDC2* [10]. None of them presented with renal disease. Since then, more and more patients with isolated liver disease (normal kidney function) or combined liver and kidney disease associated with *DCDC2* pathogenic variants have been described. With our six patients, a total of 34 patients with *DCDC2*-related hepatic ciliopathy were identified.

Based on our case series and the literature review, we conclude that the main clinical presentation of *DCDC2*-related ciliopathy is liver disease in the form of NSC. Clinically the phenotype of NSC can mimic a biliary atresia, which is difficult to exclude, especially in the neonate presenting with cholestatic jaundice with elevated GGT and acholic stools. Besides cholestasis, the main characteristic feature of DCDC2 deficiency is biliary-tract inflammation with scarring. In the ultrastructural study in liver tissue, cholangiocyte injury was accompanied by the lack of identifiable primary cilia [10]. The Authors suggested that the absence of DCDC2 may be implicated either in the formation of “cytotoxic” bile or in dysregulation of the choanocytes’ homeostatic mechanisms [10].

In patients with *DCDC2*-related cilopathy, renal phenotype was reported in a minority of them. However, based on the literature review, we would like to highlight that the absence of renal disorder at presentation does not exclude the possibility of renal involvement later in the patient’s life. Thus, NSC patients should be regularly screened for renal disease. Two main phenotypes of renal involvement were observed, NPHP and congenital anomalies of the kidney and urinary tract (CAKUT) [8,9,10,11,12,13,14,15,16,17]. The characteristic phenomenon with predominance of early and severe liver disease associated with no or mildly expressed kidney involvement in *DCDC2*-related ciliopathy needs to be fully elucidated.

The clinical phenotype of two individuals (Pt 1, 2) carrying novel disease-causing *DCDC2* variant was delineated, and collected clinical data were utilized to profile the disorder. In silico characterization of the newly identified frame shift deletion provided evidence that the consequence of this change on DCDC2 function is deleterious (12 points according ACMG classification). Loss of function is a known mechanism of *DCDC2*-related conditions (27 pathogenic LOF variants were reported in this gene). The variant affects first doublecortin functional domain—DCX1 (Figure 3). Our findings expand the molecular spectrum of pathogenic *DCDC2* variants, provide a more accurate picture of the phenotypic expression associated with molecular changes in this gene and confirm a loss of functional behaviour as the mechanism of disease.

## 5. Conclusions

The main clinical presentation of *DCDC2*-related ciliopathy is liver disease in the form of neonatal sclerosing cholangitis.The predominance of early and severe liver disease associated with no or mildly expressed kidney involvement is observed in *DCDC2*-related ciliopathy.Our findings expand the molecular spectrum of pathogenic *DCDC2* variants, provide a more accurate picture of the phenotypic expression associated with molecular changes in this gene and confirm a loss of functional behaviour as the mechanism of disease.

## Figures and Tables

**Figure 1 diagnostics-13-01917-f001:**
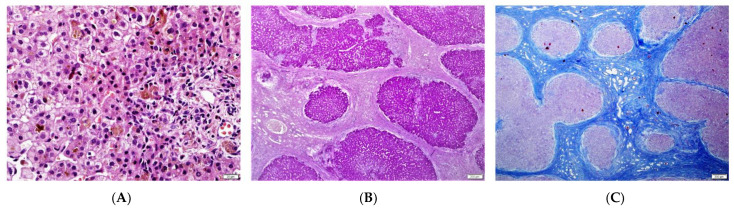
Liver histopathological findings. (**A**) Initial biopsy: severe cholestasis, giant cell transformation, portal tract with normal bile ducts and mild inflammation (H&E stain). (**B**,**C**) Liver explant, micronodular cirrhosis, ductular proliferation, cholestasis (PAS and AZAN stain).

**Figure 2 diagnostics-13-01917-f002:**
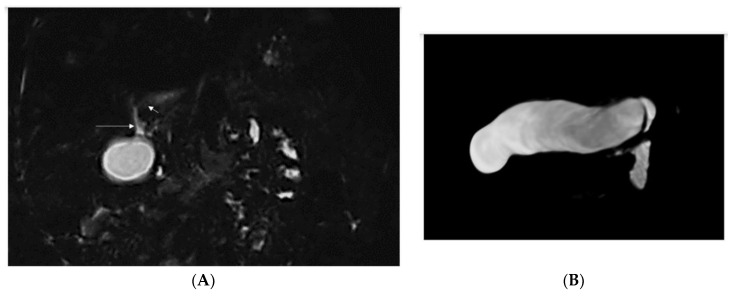
(**A**) Peripherally in segments 7 and 8 bile ducts with irregular lumen, sectionally widened up to 2.5 mm. In the remaining segments of the liver, the bile ducts are narrow, only segmentally visible, with a very irregular lumen. In the hepatic hilum, segmentally visible structures may correspond to the left bile duct and the common hepatic duct with an irregular lumen, avg. up to 1.5 mm. Right hepatic duct not visible. Long arrow = CBD, short arrow = LHD (**B**) Significantly enlarged, thin-walled gallbladder.

**Figure 3 diagnostics-13-01917-f003:**
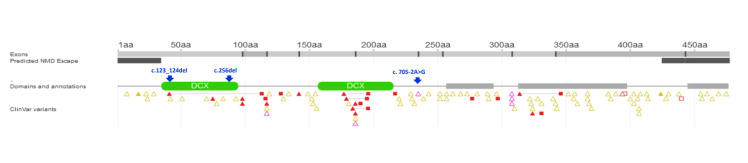
Localization of molecular variants identified in Polish patients with DCDC2-related hepatic ciliopathy (Abbreviations: DCX—Doublecortin domain; NMD—nonsense-mediated mRNA Decay).

**Table 1 diagnostics-13-01917-t001:** Clinical, pathological and molecular features of the study group.

Patient/ Gender	Origin	Age at First Presentation	Presenting Signs and Symptoms	Intrahepatic Cholangiopathy on MRCP	Liver Phenotype, Including Histology	LTx/ Age at LTx	Follow-Up	Other	Genotype
1/F	Caucasian	25th day of life	Jaundice, acholic stools, elevated GGT	No	2 m—liver biopsy: severe cholestasis, porto-portal fibrosis, moderate giant cell transformation of hepatocytes, no proliferation of bile ductules; 9 m—1st episode of oesophageal varices bleeding; 3.5 y—liver biopsy: mild cholestasis without biliary plugs, ductopenia, severe liver fibrosis; 5 y—2 episodes of oesophageal varices bleeding; 6 y—LTx with splenectomy, liver cirrhosis in the hepatic explant.	Yes/6 y	8 y	Normal kidney function	c.[123_124del];[256del], p.(Ser42Glnfs*72)/ p.(Tyr86Thrfs*17) compound heterozygote
2/F Sister of Pt 1	Caucasian	2nd day of life	Jaundice, acholic stools, elevated GGT	Yes (8 mo) See Figure 1	No liver biopsy	No	9 m	right vesico-ureteral reflux II degree, persistent urachus	c.[123_124del];[256del], p.(Ser42Glnfs*72)/ p.(Tyr86Thrfs*17) compound heterozygote
3/F	Caucasian	2 m	Jaundice, elevated GGT	n.a.	2 m—liver biopsy: mild cholestasis, portal fibrosis, focal ductular proliferation; 4 y—1st episode of oesophageal varices bleeding, ascites.	No	12.5 y	Normal kidney function	c.[123_124del];[705-2A>G] p.(Ser42Glnfs*72)/ p.? compound heterozygote
4/M Brother of Pt 3 and 5	Caucasian	2 m	Jaundice, elevated GGT	n.a.	4 m—liver biopsy: moderate cholestasis, portal fibrosis, focal ductular proliferation; 12 m—oesophageal varices in gastroscopy.	No	10 y	Normal kidney function	c.[123_124del];[705-2A>G] p.(Ser42Glnfs*72)/ p.? compound heterozygote
5/M Brother of Pt 3 and 4	Caucasian	2 m	Jaundice, elevated GGT	n.a.	No liver biopsy	No	5 y	Normal kidney function	c.[123_124del];[705-2A>G] p.(Ser42Glnfs*72)/ p.? compound heterozygote
6/M	Caucasian	3 m	Jaundice, elevated GGT	No	5 m—liver biopsy—severe cholestasis with bile plugs, ductular proliferation; 2 y—oesophageal varices in gastroscopy; 2.5 y—liver biopsy: moderate cholestasis, severe fibrosis, ductopenia; 13 y—abdominal MR—liver cirrhosis, FNH.	No	24 y	Normal kidney function	c.[123_124del];[ 123_124del] p.(Ser42Glnfs*72)/ p.(Ser42Glnfs*72) homozygote

Abbreviations: M—male; F—female; m—months; y—years; GGT—gamma-glutamyl transferase; LTx—liver transplantation; MRCP—magnetic resonance cholangiopancreatography; MR—magnetic resonance; FNH—focal nodular hyperplasia.

**Table 2 diagnostics-13-01917-t002:** Comparison of liver histopathologic features in the study group.

Patient Age at Liver Biopsy	Fibrosis	Cholestasis	Ductular Changes	Inflammation	Giant Cell Transformation
Patient 1					
2 m	porto-portal fibrosis	ductal, acinar bile plugs, severe cholestasis	no	mild portal	moderate
4.5 y	severe fibrosis	mild, hepatocellular	ductopenia	no	No
hepatic explant (6 y)	micronodular cirrhosis	mild cholestasis	diffuse ductular proliferation	mild portal	No
Patient 3					
2 m	portal fibrosis	mild, hepatocellular	focal ductular proliferation	mild portal, ductitis	No
Patient 4					
4 m	portal fibrosis	moderate hepatocellular	focal ductular proliferation	mild portal, ductitis	no
Patient 6					
5 m	periportal, bridging fibrosis	ductal, acinar bile plugs, severe cholestasis	ductular proliferation, focal DPM	mild portal	no
2.5 y	severe fibrosis	ductular bile plugs, moderate hepatocellular	ductopenia, ductular proliferation	no	no

**Table 3 diagnostics-13-01917-t003:** Clinical, pathological and molecular features of the patients reported in the literature.

Patient/ Gender	Origin/ Consanguinity	Age at First Presentation	Presenting Signs and Symptoms	Intrahepatic Cholangiopathy on ERCP/MRCP	Liver Phenotype, Including Histology	LTx/ Age at LTx	Follow-Up	Other	Genetic Result for NM_016356.5(*DCDC2*) RefSeq	References
1/n.a.	UK/Yes	n.a.	n.a.	n.a.	Hepatosplenomegaly, extensive fibrosis (11 mo) with destruction of bile ducts, bile focal duct proliferation with cholestasis	No	Died at 16 y from GI bleeding	Increased echogenicity, severe interstitial fibrosis, tubular dilation with prominent epithelial luminal budding, ESRD at 14 y	c.649A>T p.(Lys217*)/ c.649A>T p.(Lys217*)	Schueler et al., 2015 [8]
2/n.a.	Czech/No	n.a.	n.a.	n.a.	Hepatosplenomegaly, ductal plate malformation, hepatic fibrosis, scant cholestasis	Yes/2 y	current age 9 y	No renal involvement	c.123_124del, p.(Ser42Glnfs*72)/ c.349-2A>G, p.(Val117Leufs*54)	
3/F	Asian/Yes	20 wk	Jaundice, acholic stools, GGT 247 IU/L	Yes	n.a.	No/ listed	Died at 16 y	n.a.	c.649A>T, p.(Lys217*)/ c.649A>T, p.(Lys217*)	Grammatikopoulos et al., 2016 [10]
4/F	Caucasian/No	21 wk	Jaundice, ascites, splenomegaly, GI bleeding; GGT 447 IU/L	Yes	8 m—porto-portal bridging fibrosis, ductular reaction with ductal bile plugs. Hepatectomy specimen at 10 y—biliary cirrhosis, peripheral ductopenia.	Yes/10 y	12 y	n.a.	c.890T>A, p.(Leu297*)/ c.890T>A, p.(Leu297*)	
5/M	Arabic/Yes	6 wk	Jaundice, GI bleeding; GGT 711 IU/L	n.a.	8 wk—ductal plate malformation, cholestasis. Hepatectomy at 14 y—biliary cirrhosis, peripheral ductopenia.	Yes/14 y	16 y	n.a.	c.757insG, p.(Ser253Argfs*4)/ c.757insG, p.(Ser253Argfs*4)	
6/F	Caucasian/No	4 wk	Jaundice, splenomegaly; GGT 210 IU/L	Yes	Hepatectomy at 15 y—porto-portal bridging fibrosis, peripheral ductopenia, ectasia and cystic dilatation of perihilar bile ducts.	Yes/15 y	Died at 17 y	n.a.	c.529dup, p.(Ile177Asnfs*20)/ c.890T>A, p.(Leu297*)	
7/M	Caucasian/No	6 wk	Jaundice, splenomegaly; GGT 962 IU/L	Yes	9 wk—porto-portal bridging fibrosis, cholestasis, ductular proliferation with cholangiopathic features. 6 y—mild fibrosis, cholestasis, focal interlobular bile duct loss and cholangiopathic features in remaining bile ducts.	No	6 y	n.a.	c.123_124del, p.(Ser42Glnfs*72)/ c.890T>A, p.(Leu297*)	
8/M	Caucasian/No	7 wk	Jaundice, splenomegaly; GGT 365 IU/L	Yes	4 m—porto-portal bridging fibrosis, cholestasis, ductular proliferation and ductal bile plugs. Hepatectomy at 15 y—biliary cirrhosis, peripheral ductopaenia, ectasia and cystic dilatation of perihilar bile ducts.	Yes/15 y	18 y	n.a.	c.123_124del, p.(Ser42Glnfs*72)/ c.123_124del, p.(Ser42Glnfs*72)	
9/F	Caucasian/No	1 wk	Jaundice; GGT 196 IU/L	Yes	10 wk—ductal plate malformation, ductal bile plugs. 9 y—mild portal fibrosis, cholestasis, interlobular portal tract ductopaenia. Hepatectomy at 14 y—biliary cirrhosis with peripheral ductopaenia, ectasia and cystic dilatation of perihilar bile ducts.	Yes/14 y	24 y	n.a.	c.529dup, p.(Ile177Asnfs*20)/ c.890T>A, p.(Leu297*)	
10/M	n.a./Yes	n.a.	Jaundice, acholic stools	Yes	Liver histology for all 4 patients typical for NSC with early portal fibrosis, bile duct proliferation and tortuous bile ducts surrounded by fibrosis	Yes/14 y	14 y	Renal malformation—left vesico-ureteral reflux with left ureteral duplication, without ureteral dilatation. Small left kidney with right kidney hypertrophy compensation. Mild intellectual disability. Renal insufficiency after liver transplantation.	c.51G>C, p.(Lys17Asn)/ c.51G>C, p.(Lys17Asn)	Girard et al., 2016 [9]
11/M Sibling of 10	n.a./Yes	n.a.				Yes/25 y	9 y	Renal insufficiency after liver transplantation.	c.51G>C, p.(Lys17Asn)/ c.51G>C, p.(Lys17Asn)	
12/F	n.a./Yes	n.a.				Yes/6 y	n.a.	No renal disease (imaging and function) at the age of 6 y.	c.426_557del; p.(Phe142_Arg186del)/ c.426_557del; p.(Phe142_Arg186del)	
13/M Sibling of 12	n.a./Yes	n.a.				Yes/3.5 y	n.a.	4 mo—hyperechogenic left kidney at ultrasound and hypophosphatemia; normal renal function at 3 y.	c.426_557del; p.(Phe142_Arg186del)/ c.426_557del; p.(Phe142_Arg186del)	
14/M	Chinese/n.a.	2 mo	Jaundice; GGT 161-1 092 IU/L	Yes	n.a.	No	Diagnosis at 3 y 2 mo	bilateral hydronephrosis at 3 y 2 mo	c.529dup, p.(Ile177Asnfs*20)/ c.529dup, p.(Ile177Asnfs*20)	Li et al., 2018 [11]
15/M	Chinese/n.a.	9 mo	jaundice	Yes	n.a.	No	Diagnosis at 9 y 9 mo	hydrocephalus and left internal carotid artery aneurysms with vascular malformations diagnosed at 9 y 9 mo	c.529dup, p.(Ile177Asnfs*20)/ c.529dup, p.(Ile177Asnfs*20)	
16/F	African-Caribbean/Yes	n.a.	n.a.	Yes	13 y—13 years, 8 months with liver fibrosis, chronic liver failure 13 y—liver fibrosis, chronic liver failure; Hepatectomy—diffuse micronodular biliary cirrhosis, ductular proliferation	Yes (KLTx)/13 y		Nephronophthisis; Atrophic echogenic kidneys with decreased corticomedullary differentiation; Unilateral sensorineural deafness; brain imaging abnormalities	c.383C>G, p.(Ser128*)/ c.383C>G, p.(Ser128*)	Slater et al., 2019 [12]
17/M	Chinese/No	1 wk	Jaundice; GGT 247 IU/L	n.a.	early portal fibrosis and bile duct proliferation; 3 y—cirrhosis	Yes/23 y	n.a.	Abnormal creatinine, no renal biopsy	c.705-2A>G, p.?/ c.923-283_ 1023+141del, p.?	Lin et al., 2020 [14]
18/M Sibling of 17	Chinese/No	1 wk	Jaundice; GGT 102 IU/L	n.a.	n.a.	Yes/12 y	18 y	Abnormal creatinine at 18 y	c.705-2A>G, p.?/ c.923-283_1023+141del, p.?	
19/M Sibling of 17 and 18	Chinese/No	1 wk	Jaundice	n.a.	Clinical suspicion of biliary atresia; cirrhosis and liver failure at 8 m	No	Died at 8 m	n.a.	c.705-2A>G, p.?/ c.923-283_1023+141del, p.?	
20/M	Caucasian/Yes	1 wk	Jaundice, high GGT	No	Explanted liver—cirrhosis, hepatocellular and canalicular cholestasis, giant-cell change of hepatocytes	Yes/8 mo	2 y	n.a.	c.294-2A>G, p.?/ c.294-2A>G, p.?	Vogel et al., 2020 [13]
21/M	Turkish/Yes	2 wk	Jaundice, acholic stools	Yes	Liver biopsy—bilirubin stasis, cholangiolytic changes, and septal fibrosis.	No	3 y 7 m	2 y—bilateral nephronophthisis; Psychomotor delay	c.367_368del, p.(Ser123Glnfs*9)/ c.367_368del, p.(Ser123Glnfs*9)	Syryn et al., 2021 [15]
22/M	Syria/Yes	3 mo	Jaundice, hepatosplenomegaly	Yes	Liver biopsy—biliary cirrhosis; Portal hypertension with GI bleeding	Yes/2y 10 mo	n.a.	Normal renal function; Psychomotor delay, microcephaly	c.73G>A p.(Gly25Arg)/ c.73G>A p.(Gly25Arg)/	
23/M	Turkish/Yes	1 mo	Jaundice	n.a.	Liver biopsy (5 y)—congenital hepatic fibrosis	Yes (KLtx)/ 14.5 y	n.a.	Burkitt lymphoma at 11 y with renal failure development	n.a.	Duztas et al., 2022 [16]
24/F	Turkish/Yes	1 mo	Jaundice	n.a.	Liver biopsy (6 y)—cholestatic liver cirrhosis, ductopenia	Yes/6 y	n.a.	6 y—enlargement of the right kidney with a cystic mass	c.656C>G, p.(Pro2219Arg)/ c.656C>G, p.(Pro2219Arg)/	
25/F	Chinese/No	1 wk	Jaundice, elevated GGT	Yes	Giant cell changes of hepatocytes, bile plugs in hepatocytes and capillary bile ducts; ductular proliferation, cholestatic cirrhosis	No	n.a.	n.a.	c.1024-1G>T, p.?/ c.544G>A, p.(Gly182Arg)	Wei et al., 2023 [17]
26/F	Chinese/No	1 wk	Jaundice, elevated GGT	Yes	Cholestasis, liver fibrosis (stage 3), ductular proliferation, giant cell changes of hepatocytes	No	n.a.	n.a.	c.1024-1G>T, p.?/ c.544G>A, p.(Gly182Arg)	
27/M	Chinese/No	1 wk	Jaundice, elevated GGT	Yes	No liver biopsy	No	n.a.	n.a.	c.529dup, p.(Ile177Asnfs*20)/ c.529dup, p.(Ile177Asnfs*20)	
28/M	Chinese/No	1 wk	Jaundice, elevated GGT	No	No liver biopsy	No	n.a.	n.a.	c.529dup, p.(Ile177Asnfs*20)/ c.529dup, p.(Ile177Asnfs*20)	

Abbreviations: M—male; F—female; m—months; y—years; wk—weeks; n.a.—not analysed; GGT—gamma-glutamyl transferase; LTx—liver transplantation; MRCP—magnetic resonance cholangiopancreatography; ESRD—end-stage renal disease.

## Data Availability

All data generated or analysed during this study are included in this published article.

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
