# Peer review of "DCDC2-Related Ciliopathy: Report of Six Polish Patients, Novel DCDC2 Variant, and Literature Review of Reported Cases"

_diagnostics, 2023, doi:10.3390/diagnostics13111917_

Round 1

Reviewer 1 Report

In this manuscript, the authors show the clinical, pathological, and molecular features of 6 ciliopathy patients (from 3 unrelated families) with DCDC2 biallelic pathogenic variants. This study identified three different variants (one novel) in the DCDC2 gene using whole genome sequencing. Their findings expand the molecular spectrum of pathogenic DCDC2variants. However, this reviewer has the following concerns.  

Comments: 

1. On page 6 of 14, lines 3-5, not only NGS but also WES should be mentioned here, because WES plays a crucial role in this study as described in 2.3 Molecular analysis.  

2. On page 4 of 14, line 4, “Two patients (Pt 1 and 5) were biopsied at later time points (4.5 and 2.5 years, respectively)” should be “Two patients (Pt 1 and 6) were biopsied at later time points (3.5 and 2.5 years, respectively)”

3. On page 5 of 14, line 1, “One patient (Pt 1) had undergone liver transplantation (LTx) at 8 years of age.” should be “One patient (Pt 1) had undergone liver transplantation (LTx) at 6 years of age.”

4. In Table 2, “Patient 5” should be “Patient 6”.

Author Response

Dear Reviewer,

We are very grateful for Your comments.

All suggestions were corrected as advised. Please, see the revised version of the manuscript with corrections marked in red.

Reviewer 2 Report

A novel case series adequately presented. It is a case series and data pertaining to it. There is no methodology to be questioned as it is purely descriptive experience of the authors.

The following may also be discussed:

1.Wei X, Fang Y, Wang JS, Wang YZ, Zhang Y, Abuduxikuer K, Chen L. Neonatal sclerosing cholangitis with novel mutations in DCDC2 (doublecortin domain-containing protein 2) in Chinese children. Front Pediatr. 2023 Feb 3;11:1094895. doi: 10.3389/fped.2023.1094895. PMID: 36816379; PMCID: PMC9935677.

Author Response

Dear Reviewer,

We are very grateful for Your comments.

The article was cited as [17] and was included in the Table 3 as well as Discussion.

Please, the changes marked in red.